# *Kaempferia parviflora* Rhizome Extract as Potential Anti-Acne Ingredient

**DOI:** 10.3390/molecules27144401

**Published:** 2022-07-08

**Authors:** Pawee Sitthichai, Setinee Chanpirom, Tharakorn Maneerat, Rawiwan Charoensup, Thapakorn Tree-Udom, Punyawatt Pintathong, Surat Laphookhieo, Tawanun Sripisut

**Affiliations:** 1School of Cosmetic Science, Mae Fah Luang University, Chiang Rai 57100, Thailand; pawee2534@gmail.com (P.S.); setinee.cha@mfu.ac.th (S.C.); thapakorn.tre@mfu.ac.th (T.T.-U.); punyawatt.pin@mfu.ac.th (P.P.); 2Phytocosmetics and Cosmeceuticals Research Group, Mae Fah Luang University, Chiang Rai 57100, Thailand; 3School of Science, Mae Fah Luang University, Chiang Rai 57100, Thailand; wisanu.man@mfu.ac.th (T.M.); surat.lap@mfu.ac.th (S.L.); 4Center of Chemical Innovation for Sustainability (CIS), Mae Fah Luang University, Chiang Rai 57100, Thailand; 5Medicinal Plants Innovation Center of Mae Fah Luang University, Mae Fah Luang University, Chiang Rai 57100, Thailand; rawiwan.cha@mfu.ac.th; 6School of Integrative Medicine, Mae Fah Luang University, Chiang Rai 57100, Thailand

**Keywords:** *Kaempferia parviflora*, total flavonoid content, antimicrobial activity, anti-inflammatory activity, cytotoxicity, anti-acne

## Abstract

*Kaempferia parviflora* (Black ginger) is used widely in medical fields as an anti-microorganism and anti-inflammation. In this study, the aim was to evaluate the in vitro and in vivo anti-acne efficacy of black ginger extract. The results indicate that the methanol and ethanol extracts showed the highest total phenolic contents, without a significant difference, whereas the *n*-hexane extract showed the highest total flavonoid content. Nine flavones were detected using UPLC−QTOF−MS, and the ethyl acetate extract showed the highest amount of 5,7-dimethoxyflavone (DMF) according to HPLC. Antibacterial activity against *Staphylococcus aureus*, *S. epidermidis*, and *Cutibacterium acnes* was observed. All the extracts showed antimicrobial activity against C. acnes, revealing MICs in the range of 0.015 to 0.030 mg/mL, whereas the ethyl acetate extract inhibited the growth of *S. epidermidis* with a MIC of 3.84 mg/mL. In addition, the ethyl acetate extract showed the highest activity regarding nitric oxide inhibition (IC50 = 12.59 ± 0.35 µg/mL). The ethyl acetate extract was shown to be safe regarding cell viability at 0.1 mg/mL. The anti-acne efficacy was evaluated on volunteers. The volunteers were treated in two groups: one administered a 0.02% ethyl acetate extract gel-cream (n = 9) and one administered a placebo (n = 9) for 6 weeks. The group treated with the gel-cream containing the extract showed 36.52 and 52.20% decreases in acne severity index (ASI) after 4 and 6 weeks, respectively, and 18.19 and 18.54% decreases in erythema, respectively. The results suggest that *K. parviflora* could be a potent active ingredient in anti-inflammatory and anti-acne products.

## 1. Introduction

Acne vulgaris is a common inflammatory disease that appears in many adolescents. There are four factors that lead to acne vulgaris: excessive sebum production, hyperkeratinization, bacterial colonization in follicles, and inflammation [1,2]. *Cutibacterium acnes* is among the bacteria responsible for the progression of acne vulgaris. It stimulates the secretion of free radicals and inflammatory mediators, such as nitric oxide (NO), and proinflammatory cytokines, such as interleukin-1β (IL-1β), interleukin-8 (IL-8), and tumor necrosis factor (TNF-α), through toll-like receptor 2 (TLR-2). These proinflammatory cytokines can promote the development of inflammation in the skin and acne [2,3].

Acne vulgaris can be cured with many therapeutic agents such as clindamycin, erythromycin, benzyl peroxide, and salicylic acid [4,5,6,7,8,9]. However, these therapeutic agents can cause negative side effects such as skin redness and skin irritation [1,2]. Therefore, plant extracts are being studied as an alternative for treating acne [9,10].

*Kaempferia parviflora* Wall. ex Baker (black ginger) is a well-known Thai herb that is used widely in medical fields [11,12]. Several studies of black ginger rhizome extract have mentioned its pharmacological capabilities, such as curing allergies, treating obesity, and acting as an antimicrobial [13,14,15]. The black ginger rhizome contains several polymethoxyflavones that show high anti-inflammatory activity [16,17,18]. A previous study highlighted three flavonoids—5,7-dimethoxyflavone (DMF), 5,7,4′-trimethoxyflavone (TMF), and 3,5,7,3′,4′-pentamethoxyflavone (PMF)—that are major compounds in black ginger [19]. DMF, one of these major flavonoids, has been mentioned in several previous studies. As a potent anti-inflammatory agent, it can inhibit degranulation. In addition, studies have indicated its ability to inhibit tumor necrosis factor-α (TNF-α), interleukin-4 (IL-4), and monocyte chemoattractant protein-1 (MCP-1) production [18,19] as well as nitric oxide release [20].

There are currently few studies on the use of black ginger extract as an anti-inflammatory and antibacterial agent to treat acne vulgaris. Therefore, the objective of this study was to evaluate the efficacy of black ginger extract when used in acne therapy.

## 2. Results

### 2.1. Extraction Yield of K. parviflora

The yields of extraction when using various solvents were determined (Table 1). Extraction with methanol was found to provide the best extraction yield (19.69%), followed by dichloromethane (14.52%), ethyl acetate (13.18%), ethanol (13.06%), acetone (12.85%), and *n*-hexane (2.07%). *n*-Hexane showed the lowest extraction yield.

### 2.2. Total Phenolic Content

The total phenolic contents from the various solvent extractions are shown in Table 1. In the experiment, the methanol extract showed the highest total phenolic content, followed by the ethanol, ethyl acetate, acetone, dichloromethane, and *n*-hexane extracts.

### 2.3. Total Flavonoid Content

The total flavonoid contents from the various solvent extractions are shown in Table 1. The results show that the *n*-hexane extract provided the highest total flavonoid content, while the methanol extract exhibited the lowest total flavonoid content.

### 2.4. Chemical Profile According to UPLC−QTOF−MS and HPLC

#### 2.4.1. UPLC−QTOF−MS

The flavonoid constituents in the black ginger extracts were analyzed with UPLC−QTOF−MS. The flavonoid compounds along with their molecular formulas and retention times (RT) were identified (Appendix A). The compounds were identified by comparison of their mass spectrometric data obtained under positive ionization (ESI+) conditions with data in the scientific literature. The detected flavonoids with scores higher than 80 and lower than 10 ppm are reported in this study. Nine flavonoid compounds were tentatively detected in the black ginger extracts (Figure 1). 5,7-Dimethoxyflavone (DMF) was revealed to be the major compound in the methanol, ethanol, acetone, ethyl acetate, and dichloromethane extracts; 3,5,7,4′-tetramethoxyflavone was present as the major flavonoid in the *n*-hexane extract.

#### 2.4.2. HPLC Analysis

HPLC was used to determine the 5,7-dimethoxyflavone (DMF) contents in the ex-tracts. DMF was used as a marker because it is a major compound found in black ginger extracts (Figure 2). The DMF content was calculated from the DMF standard curve (R^2^ = 0.9993). The ethyl acetate extract was found to contain more DMF (119.18 ±4.02 mg/g extract) than the methanol and *n*-hexane extracts. The *n*-hexane extract presented the lowest amount of DMF (Table 2).

### 2.5. Anti-Microbial Activity

Only the ethyl acetate extract showed antimicrobial activity against *S. epidermidis*, and its MIC value was 3.84 mg/mL. All the extracts were able to inhibit *C. acnes*’ growth at low concentrations, with MICs ranging from 0.015 to 0.030 mg/mL. The MIC values of the methanol, ethanol, and *n*-hexane extracts were 0.03 mg/mL, while the acetone, ethyl acetate, and dichloromethane extracts had MIC values of 0.015 mg/mL (Table 3). However, all the extracts were inactive against *S. aureus*. None of the extracts showed bactericidal activity at 3.84 mg/mL; therefore, the MBC values were higher than 3.84 mg/mL.

### 2.6. In Vitro Anti-Inflammatory Activity


The nitric oxide (NO) inhibition was determined. The ethyl acetate extract demonstrated the highest anti-inflammatory activity, according to its IC_50_, followed by the acetone, dichloromethane, *n*-hexane, ethanol, and methanol extracts (Table 4). All the extracts had IC_50_ values lower than the IC_50_ of indomethacin. At the concentrations of the extracts and indomethacin used in the assays, the cell viability remained acceptable (≥70%). In this study, all the extracts except the *n*-hexane extract and indomethacin resulted in a significant decrease in RAW 264.7 cell viability at a concentration of 50 µg/mL.

### 2.7. Cell Viability

The cell viability of the human fibroblasts was determined by the SRB assay. The ethyl acetate extract was tested and considered to be safe, showing no cytotoxicity within a range of 0.0001 to 0.1 mg/mL (cell viability > 90%). An increase in extract concentration of up to 1 mg/mL caused the cell viability to drop drastically, to approximately 5% (Appendix A). Analysis based on the cell viability curve (R^2^ = 0.9985) showed that the cell viability was 70% when the concentration of the extract was 0.33 mg/mL.

### 2.8. Cosmetic and Stability Determination

The ethyl acetate extract was selected as the active ingredient in the cosmetic formulation in this study because of its activities in the antimicrobial and anti-inflammatory assays. An emulsion was successfully formulated into a gel-cream dosage with a light texture for skin application. The gel-cream base (CB) was basically white opaque, while the gel-cream containing the extract (CKP) was pale yellow (Appendix A). The final concentration of the ethyl acetate extract in the gel-cream formulation (CKP) was 0.02% (*w*/*w*) or 0.27 mg/mL. Both the CB and CKP, in all the storage conditions (4 °C, 45 °C, and ambient temperature), showed a slight increase in viscosity without any phase separation, flocculation, or creaming after 3 months of storage, indicating their good stability. The pH values of the freshly prepared CKP and CB were around 5.5 and then decreased steadily during storage. However, the pH values of the CKP and CB after the test were in the 5.35–5.45 range, which was acceptable.

### 2.9. Clinical Evaluation

#### 2.9.1. Closed Patch Test

The closed patch test was evaluated with 18 volunteers with acne vulgaris. After 30 min of patch removal, the results showed that, of the 18 volunteers, 3 had a mild irritant reaction to SLS (Mean irritation index; M.I.I. = 0.16). Overall, both the CB and CKP caused no skin irritation upon patch removal after 30 min (M.I.I. = 0.00). Therefore, they were classified as non-irritating products.

#### 2.9.2. Efficacy Evaluation

The clinical evaluation was conducted with volunteers who suffered from acne vulgaris, including acne spots and erythema. The volunteers were divided into two groups, the cream with extract group (CKP) and the placebo group (CB). The acne severity index (ASI) score and erythema index obtained on the first day of skin evaluation were recorded as baseline measurements. After 4 and 6 weeks, the CKP group showed 36.52 and 52.20% decreases in the ASI (*p* < 0.05), respectively. However, there was no significant change in the ASI of the CB group (Figure 3). The ASI values of both groups after 4 and 6 weeks were statistically significantly different (*p* < 0.05). Erythema presented decreases of 18.19% in the CKP group (*p* < 0.05) and 6.63% in the CB group (*p* < 0.05) after 4 weeks (Figure 4). Although the results also revealed that the significant decrease in erythema in the CB group occurred in the fourth week, the CKP group showed a greater decrease than the CB group. In Week 6, the CKP and CB groups showed slight improvements in erythema, presenting 18.54 and 11.59% decreases, respectively. Moreover, the improvement of acne spots and erythema was clearly observed in the CKP group (Figure 5) compared to the CB group.

## 3. Discussion

Flavonoids are vital phytochemical compounds that play antioxidant, anti-inflammatory, and antimicrobial clinical roles [21]. Most flavonoids in the black gin-ger rhizome are relatively low in polarity because they possess many methoxy branched structures [16,22]. Therefore, the extraction of black ginger using a medium-to-low polar solvent results in higher contents of several forms of methoxylated flavones. Crude extracts of black ginger from each solvent extraction were collected, and their percent yields were calculated. The methanol extract showed the highest yield, while the *n*-hexane extract provided the lowest yield. The high yield of the methanol extract was attributed to the number of phytochemicals such as phenolics, flavonoids, alkaloids, and terpenoids that are more soluble in methanol than in other solvents [23].

According to this study, the acetone, ethyl acetate, and dichloromethane extracts exhibited antibacterial activity against *C. acnes* with an MIC of 0.015 mg/mL. However, the ethyl acetate extract was the only one capable of inhibiting *S. epidermidis* growth (MIC = 3.84 mg/mL). A previous study reported that the ethanolic black ginger extract did not present activity against *S. epidermidis* [15], and our study showed a similar result. This might possibly be due to the higher contents of methoxyflavones present in the lower-polar solvent extract (such as ethyl acetate) [24,25], based on the results of the HPLC chromatography, which might lead to higher potency in the inhibition of *S. epidermidis* and *C. acnes*. Another previous study mentioned the antimicrobial activities against *S. aureus* and *C. acnes* of flavonoids including 5-hydroxy-7-methoxyflavone, 5-hydroxy-3,7-methoxyflavone, 5,7-dimethoxyflavone (DMF), and 5-hydroxy-3,7,4′-methoxyflavone [26]. However, in our study, not all the extracts were active against *S. aureus*. This might be due to the differences in the contents of flavones in extracts compared with the pure compounds that were used in the previous study. The MS and HPLC results represented the various amounts of methoxyflavone in the extracts from the various solvents, and our study found that, the lower the polarity of the solvent used, the higher the content of DMF obtained. Nevertheless, in our study, the *n*-hexane extract presented the lowest amount of DMF. This might indicate that DMF may have an affinity for semi-polar to low-polar solvents, but too low a polarity for the solvent used in extraction might decrease the DMF content in the extract. Because of the above reasons, the ethyl acetate extract was eventually selected as the active ingredient in the cosmetic formulation in this study.

For nitric oxide (NO) inhibition, the ethyl acetate extract possessed higher activity than the other extracts (IC_50_ = 12.59 ± 0.35 µg/mL). In a previous study, an in vitro anti-inflammatory assay with black ginger extract using RAW 264.7 cell lines was performed. Organic solvent extracts showed higher activity in NO suppression than the water extract [17]. In this study, a medium-polarity solvent (such as ethyl acetate) showed significantly higher anti-inflammatory activity than higher-polarity solvents (such as methanol and ethanol). NO is a free radical generated from inducible nitric oxide synthase (iNOS). A previous study showed that nitrate and nitrite levels were high in acne patients [27]. Thus, NO production could be a factor in acne vulgaris. DMF and other flavones from black ginger extract were isolated in a previous study, and these flavones were shown to inhibit iNOS and reduce NO production in RAW 264.7 cells [28]. The study also reported that DMF in the extract could suppress iNOS expression in *C. acnes*-stimulated human keratinocyte cells (HaCaTs) [28]. Thus, this could lead to the attenuation of NO secretion, which ultimately reduces inflammation in acne.

In our study of the efficacy of the ethyl acetate black ginger extract, a group of volunteers using CKP exhibited a decrease in acne spots and inflammation. The inflammatory mechanism of acne can be triggered by pathogenetic bacteria including *C. acnes*, *S. epidermidis,* and *S. aureus* [2,3]. *C. acnes*, a predominant bacterium in hair follicles, is a factor in acne vulgaris and inflammation that triggers proinflammatory cytokines, such as IL-8, IL-12, IL-1α, IL-1β, and TNF-α, in human keratinocytes, sebocytes, and microphages through toll-like receptor-2 (TLR-2), which induces the NF-κB pathway. This eventually causes inflammation in acne [2,29,30,31]. Moreover, *C. acnes* produces porphyrin, which triggers reactive oxygen species (ROS). This also promotes inflammation in keratinocytes [32,33]. Flavones have been reported to possess antibacterial activities via different mechanisms. Several studies have mentioned that they can inhibit the bacterial cell wall’s adhesion, obstructing bacterial growth. They can also inhibit bacterial enzymes and act as bacterial efflux pump inhibitors, leading to a decrease in biofilm formation [34,35]. A previous study reported that a flavonoid, 5-hydroxy-7-methoxyflavone, showed anti-biofilm activity against *S. aureus* [36]. Thus, DMF could possess the same mechanism, owing to the similarity of these compounds. In addition, the downregulation of NF-κB production by black ginger extract through the blockade of phosphorylation that leads to the induction of the NF-κB signaling pathway was also mentioned in previous studies [27,32,37]. For the aforementioned reasons, the gel-cream containing black ginger extract gradually reduced the severity of acne vulgaris and inflamed spots in the volunteers.

However, additional determinations, such as measurements of the sebum and porphyrin contents, are required for more detailed knowledge. In addition, varying the concentrations of the extract in the formulation to find the minimum concentration that produces effective results will be necessary to reduce the costs in industry.

## 4. Materials and Methods

### 4.1. Chemicals

Gallic acid monohydrate, quercetin, and 5,7-dimethoxyflavone (DMF) were purchased from Sigma Aldrich (St. Louis, MI, USA). All the media, including Mueller–Hinton broth, tryptic soy broth, and brain heart infusion broth, were purchased from Himedia (Mumbai, India). Folin–Ciocalteu reagent was purchased from LOBA Chemie (Mumbai India). All the solvents used in this study were analytical grade.

### 4.2. Plant Material

Dried chopped black ginger rhizomes were purchased in August 2018 from Petchabun Province, Thailand. The rhizomes were ground into fine powder and stored at −20 °C until used.

### 4.3. Preparation of K. parviflora Extracts

The rhizome powder of *K. parviflora* (100 g) was macerated separately with various solvents—methanol, ethanol, acetone, ethyl acetate, dichloromethane, and *n*-hexane (ratios of 1:5, *w*/*v*)—for 24 h at a 200 rpm shaking speed at ambient temperature. After that, each sample was filtered by vacuum filtration with filter paper (Whatman no. 1) to separate the residue from the solution. The remaining plant residue was extracted two times, and the solvents were removed by using a rotary evaporator. The black ginger extracts were placed overnight in a desiccator. The yields of the extracts were separately calculated, and the obtained extracts were stored at 4 °C until used.

### 4.4. Determining Total Phenolic Content

The Folin–Ciocalteu method was conducted using a 96-well plate, with modifications based on Mahboubi et al. [38]. A total of 20 µL of sample solution was added into a 96-well plate. Then, 100 µL of Folin–Ciocalteu reagent (10-fold diluted with deionized water) was added before the reaction was left for 3 min, which was followed by adding 80 µL of Na_2_CO_3_ solution (7.5% *w*/*v*). The reaction was shaken and kept in the dark for an hour at room temperature. The absorbance of the reaction was measured at 765 nm. Gallic acid was used as a standard. The experiment was repeated in triplicate, and the results are expressed as the gallic acid equivalent per gram of extract (mg GAE/g of plant extract).

### 4.5. Determining Total Flavonoid Content

The total flavonoid content was measured by an aluminum chloride colorimetric assay, which was modified from Ordonez et al. [39], using a 96-well plate. A 100 µL aliquot of sample solution was mixed with the same volume of 2% *w*/*v* AlCl_3_ solution (dissolved in 80% ethanol). The reaction was placed in the dark at room temperature for 1 h. The absorbance of the resulting solution was measured at 420 nm. Quercetin was used as a standard. The experiment was performed in triplicate, and the results are expressed as the quercetin equivalent per gram of extract (mg QE/g of plant extract).

### 4.6. Chemical Profiling by Chromatography Techniques

#### 4.6.1. UPLC–QTOF–MS Analysis

LC–MS/MS was performed using an Agilent LC-QTOF 6500 system with an Agilent ZORBAX Eclipse XDB column–C18 (2.1 × 50 mm, 1.7 µm). For the mobile phases, water with 0.1% formic acid (mobile phase A) and acetonitrile with 0.1% formic acid (mobile phase B) were used as the eluents in a gradient mode. The column was eluted at a flow rate of 1 mL/min using a linear gradient as follows: 5% (B) at 0–1 min, 5–17% (B) at 1–10 min, 17% (B) at 10–13 min, 17–100% (B) at 13–20 min, 100% (B) at 20–22 min, 100–5% (B) at 22–25 min, and 5% (B) at 25–26 min. The injection volume was 20 μL, and the column’s temperature was maintained at 30 °C. The UPLC system was coupled to a QTOF mass spectrometer (6500 series; Model-G6545B) (Agilent Technologies, Santa Clara, CA, USA) equipped with an ESI Spray source. The parameters for analysis were set using positive ion mode, with spectra acquired over a mass range from m/z 120 to 1000. The MS data were processed using the MassHunter v B.08.00, Rapid Control V 2.9 software (accessed on 8 April 2020), which provided a list of possible elemental formulas by integration with libraries. The Agilent MassHunter Qualitative Analysis Software version 6.00 was used for the initial processing of the LC/MS data. The compounds were revealed using the Molecular Feature Extractor (MFE) tool in the software. The Agilent Metlin Metabolite Personal Compound Database and Library (PCDL) version 5.0 was used to identify the compounds based on the matching of the MS/MS spectra.

#### 4.6.2. HPLC Analysis

For the HPLC analysis, an InfinityLab Poroshell 120 EC-C18 (4.6 × 150 mm) column (Agilent Technologies, Santa Clara, CA, USA) was used. With isocratic elution, the mobile phases consisted of 0.5% (*v*/*v*) formic acid in water (mobile phase A) and methanol (mobile phase B), with a ratio 30:70%, *v*/*v*, for 25 min. The flow rate was 0.5 mL/min, and the column temperature was maintained at 30 °C. The injection volume was 10 µL, and the detection wavelength was set at 265 nm

### 4.7. Anti-Microbial Activity

#### 4.7.1. Microorganisms

The tested organisms included *Staphylococcus aureus* TISTR746, *Staphylococcus epidermidis* TISTR2141, and *Cutibacterium acnes* DIMST14916. *S. aureus* and *S. epidermidis* were cultured using tryptic soy agar (TSA) under aerobic conditions, while *C. acnes* was cultured using brain heart infusion agar (BHI agar) under anaerobic conditions.

#### 4.7.2. Determination of Minimum Inhibition Concentration (MIC)

The broth microdilution method mainly followed Wiegand et al. [40] with some modifications using a 96-well plate. *S. aureus* and *S. epidermidis* were inoculated by picking colonies and transferring them into tubes containing nutrient broth (NB), which were shaken using an incubator shaker (Shel Lab, Cornelius, OR, USA) for 24 h at 37 °C. For *C. acnes*, the colony was used to inoculate brain heart infusion (BHI) and incubated for 72 h under anaerobic conditions. The extract samples were dissolved in acetone and separately diluted with each broth (NB and BHI) to make a 12.8% (*v*/*v*) acetone solution. Then, the sample solutions were diluted in a twofold serial dilution in sterile 96-well plates (0.015–7.68 mg/mL). The final bacterial suspensions were prepared by the 200-fold dilution of the 0.5 McFarland standard using 0.85% (*w*/*v*) sterile saline and its addition to the wells containing the sample solutions. The plates containing *S. aureus* and *S. epidermidis* were incubated at 37 °C for 24 h under aerobic conditions, while the plates containing *C. acnes* were incubated at 37 °C for 72 h under anaerobic conditions. After incubation, 10 µL of resazurin solution (0.18% *w*/*v*) was added into all the wells, and the plates were incubated at room temperature for a further 3 h. The MIC was determined from the change in color. Vancomycin and gentamicin were used as positive controls, dissolved separately in sterile water (0.00025–0.128 mg/mL). Each strain suspension was mixed with 12.8% (*v*/*v*) acetone solution to be used as a control.

#### 4.7.3. Determination of Minimum Bactericidal Concentration (MBC)

The method followed Wiegand et al. [40]. As for the MIC assay, 10 µL of mixed suspension in the wells showing an absence of bacterial growth (no change in the color of the resazurin from dark blue to pink) was transferred and streaked on a Mueller–Hinton agar (MHA) plate, which was incubated for 18 h at 37 °C for the aerobic microorganisms. For the anaerobic microorganisms, BHI agar was used instead of MHA, and the incubation was performed for 72 h at 37 °C. The lowest concentration of extract that completely killed the bacteria was determined as the MBC.

### 4.8. In Vitro Anti-Inflammatory Activity

#### 4.8.1. RAW 264.7 Cell Viability

The cytotoxicity toward RAW 264.7 cells was determined using the resazurin assay. The method was conducted with reference to Riss et al. [41], with slight modifications. Briefly, resazurin solution (0.5 mg/mL) was added to each well of the 96-well plate (10 µL), and the plate was incubated at 37 °C for 4 h. At the end of the incubation time, the fluorescence intensity of the samples was determined at 560 nm for excitation and 590 nm for emission.

#### 4.8.2. Nitric Oxide Inhibition Assay

The inhibition of nitric oxide production was determined according to Dzoyem et al. [42]. Briefly, RAW 264.7 cells were cultured in Dulbecco’s Modified Eagle Medium (DMEM) supplemented with 10% fetal bovine serum (FBS), 2 mM L-glutamine, and 100 U/mL penicillin/streptomycin. Samples (stock solution) were dissolved in DMSO and diluted to 0.0031 to 0.05 mg/mL to make 0.5% (*v*/*v*) DMSO in the culture medium. The cells were seeded in a 96-well plate (4 × 10^5^ cells/well) and stimulated with lipopolysaccharide (LPS) for 1 h. After 1 h of incubation, the cells were incubated with various concentrations of samples for 24 h. After the incubation time, 50 µL of supernatant was mixed with 50 µL of Griess reagent (1% sulfanilamide, 0.1% naphthyl ethylene diamine dihydrochloride, and 2.5% phosphoric acid) and incubated for 10 min. The nitric oxide was detected at 540 nm using a microplate reader; Indomethacin was used as a positive control, dissolved in 5% DMSO solution (*v*/*v*), and diluted in a twofold serial dilution (0.0062 to 0.1 mg/mL).

### 4.9. Cell Viability

The cell viability was assessed by a Sulforhodamine B (SRB) colorimetric assay, and human fibroblast cells were used as the cell lines in the study. The ethyl acetate extract was chosen because of its activities in the antimicrobial and anti-inflammatory assays. The cell viability assay was performed according to Vichai and Kirtikara [43]. To prepare the sample solution, briefly, the extract was dissolved in DMEM with 10% fetal bovine serum (FBS) and 1% penicillin/streptomycin containing 0.5% (*v*/*v*) polysorbate 80. The mixed solution was filtered through a membrane (0.2 microns). The cells were treated with various concentrations of the samples, and the plate was left for 72 h. Cell fixation and staining were then performed. The plate was then washed to eliminate excess dye, and the bound dye in the plate was solubilized by using tris buffer (pH 10.5) before the absorbance was measured. The absorbance representing the cell viability was monitored at 510 nm. Cell viability of 70% and above was considered to represent a lack of cytotoxicity [44]. Sodium lauryl sulfate (SLS) was used as a positive control.

### 4.10. Cosmetic Formulation

The gel-cream dosage was prepared as the cosmetic formulation in this study. Briefly, the aqueous and oil phases were prepared separately. The aqueous phase consisted of deionized water, disodium EDTA, butylene glycol, ethoxydiglycol, and Aristoflex AVC (ammonium acryloyldimethyltaurate/VP copolymer). The oil phase contained polysorbate 20, isononyl isononanoate, isododecane, and Viscolam AT 100P (sodium polyacryloyldimethyl taurate, hydrogenated polydecene, and Trideceth-10). The water phase was steadily added into the oil phase with the use of a homogenizer to mix the formulation well. Both Liquid Germall^TM^ Plus (propylene glycol and diazolidinyl urea and iodopropynyl butylcarbamate) and Mild Preserved Eco^TM^ (1,3-propanediol and ethylhexylglycerin) were added as preservatives. Adjustment of pH in gel-cream to approximately 5.5 was done by adding triethanolamine. The black ginger ethyl acetate extract was dissolved in co-solvents (ethanol: ethoxydiglycol: polysorbate 20), and the solution was added into the gel-cream to obtain the gel-cream containing extract (CKP). For the gel-cream base (CB), plain co-solvents were added into the gel-cream instead. The final concentration of the extract in gel-cream was 0.02% (*w*/*w*) or 0.27 mg/mL.

### 4.11. Stability Determination

#### 4.11.1. Accelerated Stability Test

An accelerated stability test was performed to determine the stability of the formulations according to Venter et al. [45] with slight modifications. Both the CKP and CB were placed separately under three different storage conditions (45 °C, 4 °C, and ambient temperature). The viscosity and pH of both formulations were initially measured at baseline before the tests (Week 0), and we continued to measure them at Weeks 4, 8, and 12 (3 months overall) to determine the changes in their viscosities and pH values.

#### 4.11.2. Viscosity Measurement

A viscometer (Brookfield, Middleborough, MA, USA) was used to determine the viscosity changes in the formulations during storage. A no. 6 needle was used to measure the viscosities of the formulations after 30 s at 100 rpm. The experiment was performed in triplicate.

#### 4.11.3. pH Measurement

The pH was measured to determine the pH changes in the formulations during storage using a pH meter. The measurements were conducted in triplicate.

### 4.12. Clinical Evaluation

#### 4.12.1. Inclusion Criteria

Healthy Thai volunteers aged between 20 and 40 years old with mild-to-moderate acne were enrolled in this study. The classification of acne severity mainly referred to Hayashi et al. [46]. All the recruited subjects were informed about the study both in writing and verbally, and they signed a written consent form, which was approved by the ethical committee of the Mae Fah Luang University prior to enrollment (EC-20021-17). The human volunteer study was governed by the Declaration of Helsinki.

#### 4.12.2. Irritation Test

A closed patch test was conducted on 18 subjects to determine the irritation responses as a preliminary test. Four different plates including CKP, CB, 0.5% *w*/*v* sodium lauryl sulfate (SLS) solution (positive control), and deionized water (negative control) were applied to the skin on the anterior upper arm of volunteers for 24 h. Observation was undertaken after Finn chamber^®^ (8 mm, SmartPractice, Phoenix, AZ, USA) removal for 30 min. As demonstrated in Table 5 and Table 6, the mean irritation index (M.I.I.) was calculated using the following equation based on Baldisserotto et al. (Table 6) [47].
M.I.I. = (overall score of erythema and edema) ÷ (number of volunteers).(1)

#### 4.12.3. Efficacy Evaluation

Eighteen healthy Thai volunteers between the ages of 20 and 40 years old were included in this single-blind study. All the subjects were allergy free for 1 week and had not used steroids and topical products for inflammatory and anti-acne treatment for 4 weeks prior to the study. No other facial products except sunscreen were allowed during the study. The volunteers were divided into two groups and randomly received different formulations of gel-cream. The first group received the CKP, and the second group received the CB. The volunteers were directed to apply the formulations to their faces twice daily after face cleansing in the morning and evening for 6 weeks. Before skin measurement, the volunteers washed their faces and rested in the room for 30 min under constant environmental conditions (22 ± 2 °C and 55 ± 5% relative humidity). The acne spots and erythema conditions were analyzed using the acne severity index (ASI) score system [48], using the Mexameter^®^ (MX18, Courage Khaazaka electronic, Cologne, Germany) and taking photos at baseline and 2, 4, and 6 weeks after the product’s application. The experimental protocol followed Choi et al. [49], with minor modifications. The ASI score was calculated using the following equation:ASI = (no. of pustule × 2) + (no. of papules) + (no. of comedones ÷ 4).(2)

### 4.13. Statistical Analysis

All data except for clinical efficacy were presented as mean ± standard deviation (SD), and all analyses were performed in triplicate. The data obtained from in vitro and cellular assays were compared and analyzed using one-way ANOVA (analysis of variance) to analyze the data for statistically significant differences, whereas the data from clinical efficacy test were analyzed by using the paired *t*-test and expressed as mean ± standard error of mean (SEM). Statistical significance was tested at a 5% level of significance; therefore, *p* < 0.05 indicated statistically significant differences between the compared values. The analyses were performed using the SPSS software ver. 22.0 (IBM, Armonk, NY, USA).

## 5. Conclusions

The study showed different results for the black ginger extracts with the different solvent extractions (regarding the TPC, TFC, antimicrobial activity, anti-inflammatory activity, and chromatography data). The methanol extract contained the highest total phenolic content, while the *n*-hexane extract showed the highest total flavonoid content. The ethyl acetate extract inhibited the growth of both *S. epidermidis* and *C. acnes* as well as showing the greatest activity in nitric oxide inhibition. The formulated gel-cream containing the ethyl acetate extract demonstrated good stability, without significant changes in viscosity and pH. The volunteers who used the gel-cream containing the ethyl acetate extract showed significant reductions in inflammation from acne vulgaris and erythema in 6 weeks. Therefore, the implication of the results suggests that the black ginger extract might be used as an active ingredient in anti-inflammatory and anti-acne products.

## Figures and Tables

**Figure 1 molecules-27-04401-f001:**
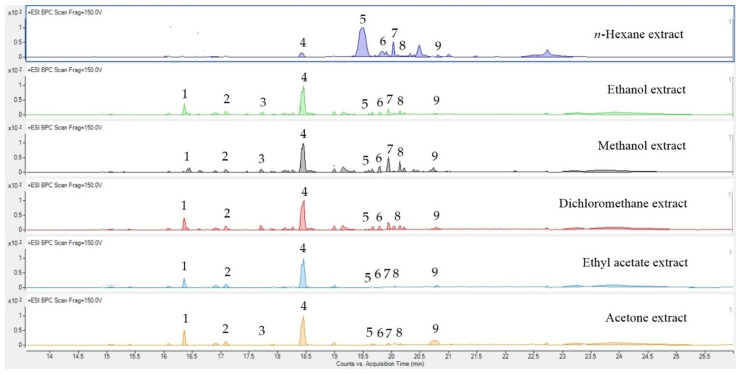
UPLC-QTOF-MS chromatograms of *K. parviflora* extracted using different solvents; 5,7,4′-Trimethoxyflavone (**1**), 3,5,7-Trimethoxyflavone (**2**), 5-Hydroxy-7,4′-dimethoxyflavone (**3**), 5,7-Dimethoxyflavone (**4**), 3,5,7,4′-Tetramethoxyflavone (**5**), 5-Hydroxy-3,7,3′,4′-tetramethoxyflavone (**6**), 5-Hydroxy-7-methoxyflavone (**7**), 5-Hydroxy-3,7-dimethoxyflavone (**8**), and 5-Hydroxy-3,7,4′-trimethoxyflavone (**9**).

**Figure 2 molecules-27-04401-f002:**
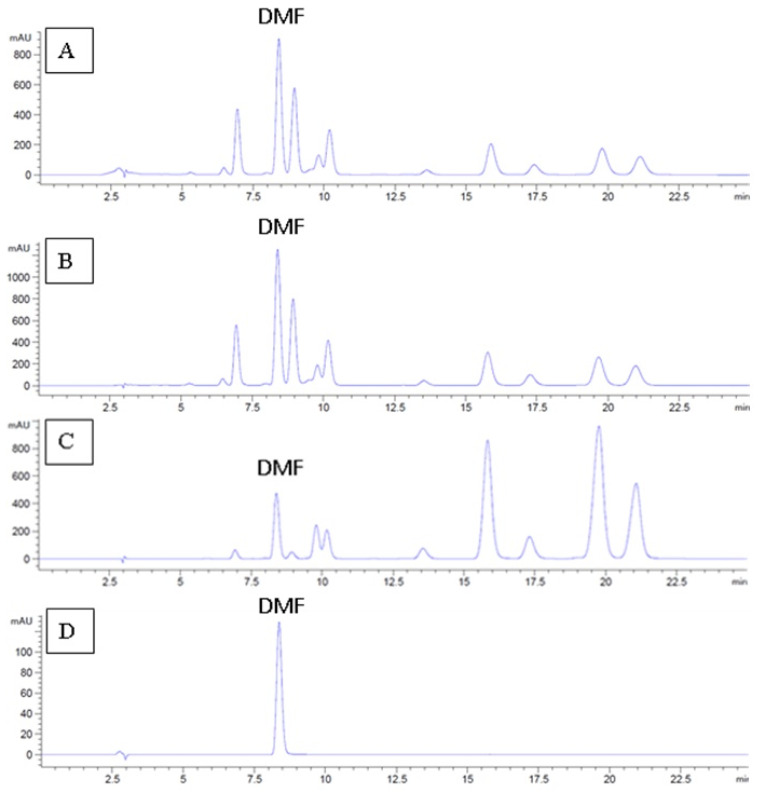
HPLC chromatogram and DMF peak for each *K**. parviflora* extract; methanol extract (**A**), ethyl acetate extract (**B**), *n*-hexane extract (**C**), and standard DMF compound (**D**).

**Figure 3 molecules-27-04401-f003:**
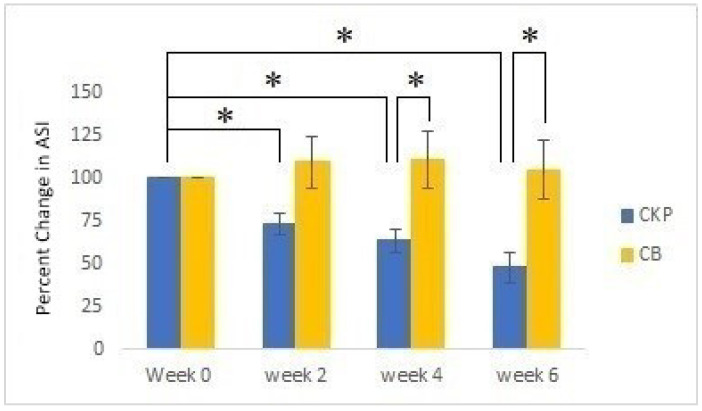
Percent changes in the acne severity index (ASI) scores between Groups A (CKP) and B (CB; control) in Weeks 0, 2, 4, and 6; * represents a significant difference (*p* < 0.05). Values represent means ± SEM.

**Figure 4 molecules-27-04401-f004:**
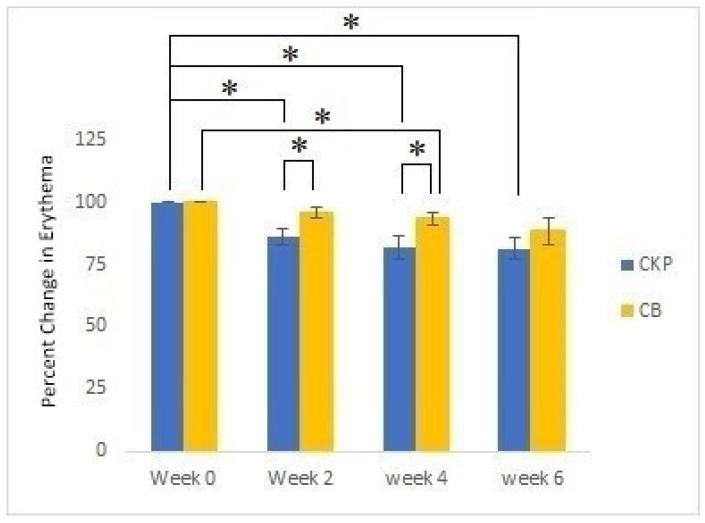
Percent changes in the erythema index between Groups A (CKP) and B (CB; control) in Weeks 0, 2, 4, and 6; * represents a significant difference (*p* < 0.05). Values represent means ± SEM.

**Figure 5 molecules-27-04401-f005:**
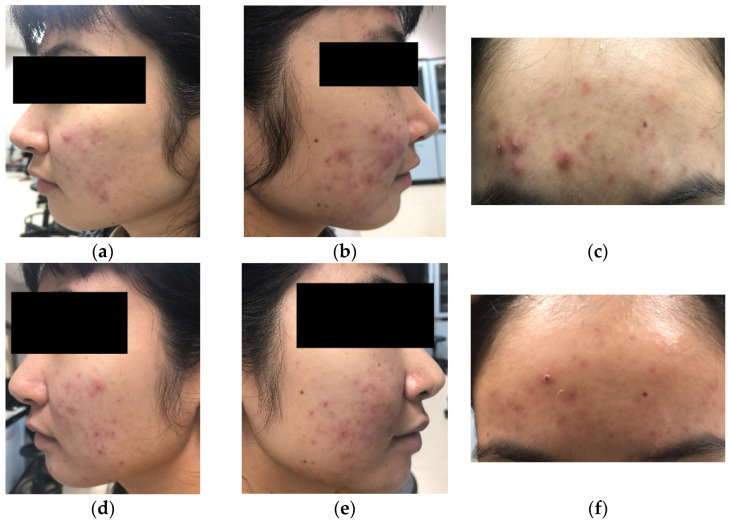
One representative acne volunteer in Week 0 (**a**–**c**) and Week 6 (**d**–**f**).

**Table 1 molecules-27-04401-t001:** Extraction yield, total phenolic, and flavonoid content of each *K. parviflora* ex-tract.

Conditions	Yield (%)	TPC (GAE mg/g Extract)	TFC (QE mg/g Extract)
Methanol ext.	19.69	58.45 ± 2.85 ^a^	47.92 ± 0.92 ^d^
Ethanol ext.	13.06	52.28 ± 3.85 ^a, b^	57.56 ± 3.33 ^c^
Acetone ext.	12.85	42.05 ± 4.16 ^c^	62.22 ± 2.25 ^b, c^
Ethyl acetate ext.	13.18	45.46 ± 4.42 ^b, c^	62.90 ± 0.97 ^b, c^
Dichloromethane ext.	14.52	36.69 ± 0.53 ^c, d^	64.82 ± 3.15 ^b^
*n*-Hexane ext.	2.07	32.31 ± 0.43 ^d^	127.09 ± 1.00 ^a^

TPC: total phenolic content; TFC: total flavonoid content; different letters (^a^, ^b^, ^c^ and ^d^) indicate statistical significance at *p* < 0.05. Results are expressed as means ± SDs.

**Table 2 molecules-27-04401-t002:** HPLC analysis of DMF in each *K. parviflora* extract.

Sample	DMF Amount (mg/g Extract)
Methanol ext.	92.72 ± 3.21 ^b^
Ethyl acetate ext.	119.18 ± 4.02 ^a^
*n*-Hexane ext.	48.46 ± 1.70 ^c^

Different letters (^a^, ^b^ and ^c^) indicate statistical significance at *p* < 0.05. Results are expressed as means ± SDs.

**Table 3 molecules-27-04401-t003:** Antimicrobial activities of *K. parviflora* extracts.

Samples	MIC (mg/mL)		
	*S. aureus*	*S. epidermidis*	*C. acnes*
Methanol ext.	>3.84	>3.84	0.03
Ethanol ext.	>3.84	>3.84	0.03
Acetone ext.	>3.84	>3.84	0.015
Ethyl acetate ext.	>3.84	3.84	0.015
Dichloromethane ext.	>3.84	>3.84	0.015
*n*-Hexane ext.	>3.84	>3.84	0.03
Vancomycin	0.0005	0.001	0.001
Gentamycin	<0.0001	<0.0001	0.004

**Table 4 molecules-27-04401-t004:** Nitric oxide inhibition by *K. parviflora* extracts.

Samples	IC_50_ (µg/mL)
Methanol ext.	20.02 ± 0.20 ^c^
Ethanol ext.	19.51 ± 0.62 ^c^
Acetone ext.	13.47 ± 0.17 ^b^
Ethyl acetate ext.	12.59 ± 0.35 ^a^
Dichloromethane ext.	13.91 ± 0.15 ^b^
*n*-Hexane ext.	13.95 ± 0.21 ^b^
Indomethacin	23.13 ± 1.51 ^d^

Different letters (^a^, ^b^, ^c^ and ^d^) indicate statistical significance at *p* < 0.05. Results are expressed as means ± SDs.

**Table 5 molecules-27-04401-t005:** Grading criteria for skin erythema and edema.

Grade	Erythema	Edema
0	Negative reaction	Negative reaction
+1	Light erythema	Very slight edema
+2	Clearly visible erythema	Slight edema
+3	Moderate erythema	Moderate edema
+4	Intense erythema	Strong edema

**Table 6 molecules-27-04401-t006:** Mean irritation index (M.I.I.) interpretation.

M.I.I. Value	Irritancy
M.I.I. < 0.50	No irritation
0.50 ≤ M.I.I. < 2.00	Mild irritation
2.00 ≤ M.I.I. < 5.00	Moderate irritation
M.I.I. > 5.00	Strong irritation

## Data Availability

Data are contained within the article or Appendix A.

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
