# Peer review of "Kaempferia parviflora Rhizome Extract as Potential Anti-Acne Ingredient"

_molecules, 2022, doi:10.3390/molecules27144401_

Round 1
Reviewer 1 Report
The manuscript molecules-1788418 brings novel insights into the chemical characterization of Kaempferia parviflora rhizomes and their clinical application as anti-acne agents.
The manuscript needs several improvements before publication in Molecules, as follows:
In vitro Antimicrobial study: Please specify whether you used reference strains or clinical isolates for the antimicrobial susceptibility testing. For clinical isolates, please add a brief description of the isolates; origin, disease caused by, its antibiogram susceptibility and resistance. Moreover, the final concentration of acetone in the culture medium is extremely high! (12.8%)
Anti-inflammatory assay: the final concentration of DMSO in the culture medium is extremely high! (5%)
Cream formulation: please specify the concentration of the extract found in the obtained formulation. There is no reference to this fact, which is of most importance to link the in vitro testing to the clinical study.
Clinical study: only 18 subjects were enrolled; I think the number of patients is too small for the clinical study to obtain statistically significant data.
The manuscript needs to be checked by a native English speaker.
Author Response
|
Comment |
Edit data and note |
Reviewer 1
|
In vitro Antimicrobial study: Please specify whether you used reference strains or clinical isolates for the antimicrobial susceptibility testing. For clinical isolates, please add a brief description of the isolates; origin, disease caused by, its antibiogram susceptibility and resistance. Moreover, the final concentration of acetone in the culture medium is extremely high! (12.8%) |
(L324-325): The reference strains are mentioned in 4.7.1. Microorganisms. S. aureus and S. epidermidis were received from the Thailand institute of scientific and technological research. C. acnes was received from Department of medical sciences, National institute of health of Thailand.
(L335-336 and 345-346): This concentration of acetone (12.8% v/v) is initial concentration before adding bacterial suspension into the first well. The final concentration after adding bacterial suspension into the well is 6.4% (v/v). Bacterial growth was observed in the growth control containing 6.4% (v/v) as the final concentration. Therefore, 6.4% (v/v) acetone is considered to be safe [1-3]. |
Anti-inflammatory assay: the final concentration of DMSO in the culture medium is extremely high! (5%) |
(L366-367 and L373-375): DMSO and 5% DMSO (v/v) were used as solvents for dissolving extracts (samples) and indomethacin, respectively. After serial dilutions were performed, the final concentrations of DMSO were 0.5% (v/v) and lower. Therefore, 0.5% (v/v) is considered to be safe [4-5]. |
|
Cream formulation: please specify the concentration of the extract found in the obtained formulation. There is no reference to this fact, which is of most importance to link the in vitro testing to the clinical study. |
(L405-406): The concentration of active ingredient in cosmetic formulation is commonly presented in weight-by-weight percent (0.02%). The concentration of extract in formulation in mg/mL unit was 0.27 mg/mL.
(L150 and L405-406): The concentration in mg/mL unit has been added.
|
|
The manuscript needs to be checked by a native English speaker. |
English was proved by MDPI English Editing (ID: English-44769) |
References
- Eloff, J. A sensitive and quick microplate method to determine the minimal inhibitory concentration of plant extracts for bacteria.Planta Med. 1998, 64, 711–713. doi:10.1055/s-2006-957563.
- Eloff, J. Which extractant should be used for the screening and isolation of antimicrobial components from plants? J. Ethnopharmacol. 1998, 60, 1–8. doi:10.1016/s0378-8741(97)00123-2.
- Poljuha, D.; Sladonja, B.; Šola, I.; Šenica, M.; Uzelac, M.; Veberič, R.; Hudina, M.; Famuyide, I.M.; Eloff, J.N.; Mikulic-Petkovsek, M. LC–DAD–MS phenolic characterisation of six invasive plant species in Croatia and determination of their antimicrobial and cytotoxic aactivity.Plants 2022, 11, 596. doi:10.3390/plants11050596
- Dzoyem, J.P.; Donfack, A.R.; Tane, P.; McGaw, L. J.; Eloff, J.N. Inhibition of nitric oxide production in LPS-stimulated RAW 264.7 macrophages and 15-LOX activity by anthraquinones from Pentas Schimperi. Planta Med. 2016, 82, 1246–1251. doi:10.1055/s-0042-104417.
- Ilieva, Y.; Dimitrova, L.; Zaharieva, M.M.; Kaleva, M.; Alov, P.; Tsakovska, I.; Pencheva, T.; Pencheva-El Tibi, I.; Najdenski, H.; Pajeva, I. Cytotoxicity and Microbicidal Activity of Commonly Used Organic Solvents: A Comparative Study and Application to a Standardized Extract from Vaccinium macrocarpon.Toxics 2021, 9, 92. doi:10.3390/toxics9050092
Reviewer 2 Report
L84 - correct to TPC
L368 what was the concentration of DMSO?
L475-476 "Therefore, black ginger extract can be used as an active ingredient in anti-inflammatory and anti-acne products." In my opinion the authors should concluded their results more cautiusly. Presented results of the clinical part of the study are not sufficient yet to assume such use of the extract.
Author Response
|
Comment |
Edit data |
Reviewer 2 |
L84 - correct to TPC |
(L84): The “PC” has been corrected to TPC.
|
L368 what was the concentration of DMSO? |
(L366-367): The concentration of DMSO for dissolving the black ginger extracts has been added. We used pure DMSO to dissolve the extracts (10 mg/mL). Then, it was diluted with water to decrease the concentration of DMSO to 0.5% (v/v) and below (from 0.05 mg/mL to 0.0031 mg/mL). Therefore, cytotoxicity was not observed [4-5].
|
|
L475-476 "Therefore, black ginger extract can be used as an active ingredient in anti-inflammatory and anti-acne products." In my opinion the authors should concluded their results more cautiously. Presented results of the clinical part of the study are not sufficient yet to assume such use of the extract. |
(L474-475): The sentence has been edited. |

Reviewer 3 Report
The authors explained the matter of active ingredient concentration-which is a bit higher in the cream than the concentration used in a cellular assay. However, there is still a problem. The active ingredient concentration used in the assay and in the cream products tested on volunteers is cytotoxic (cytotoxicity started from 0.1 mg/mL and the concentration in the cream was 0.27 mg/mL). How is that safe?
Author Response
|
Comment |
Edit data |
Reviewer 3 |
The authors explained the matter of active ingredient concentration-which is a bit higher in the cream than the concentration used in a cellular assay. However, there is still a problem. The active ingredient concentration used in the assay and in the cream products tested on volunteers is cytotoxic (cytotoxicity started from 0.1 mg/mL and the concentration in the cream was 0.27 mg/mL). How is that safe? |
The amount of plant extracts in formulation is normally higher than the effective doses obtained from cellular assay. This is because the active compounds in topical formulation do not easily penetrate the skin [6-8]
(L141-142): 70% of Cell viability and above was considered to represent a lack of cytotoxicity [9]. In this study, cell viability curve (R2 = 0.9985) showed that the cell viability was 70% when the concentration of the extract was 0.33 mg/mL. Thus, gel-cream containing the black ginger extract at 0.27 mg/mL is considered to be safe.
In addition, skin irritation test showed no sign of irritation or allergic reaction with gel-cream containing black ginger extract by closed patch test. |
References
- Nakyai, W.; Pabuprapap, W.; Sroimee, W.; Ajavakom, V.; Yingyongnarongkul, B. E.; Suksamrarn, A. Anti-acne vulgaris potential of the ethanolic extract of Mesua ferrea L. flowers. Cosmetics 2021, 8, 107. doi:10.3390/cosmetics8040107.
- Shin, S.; Lee, J.-A.; Son, D.; Park, D.; Jung, E. Anti-skin-aging activity of a standardized extract from Panax ginsengleaves in vitro and in human volunteer. Cosmetics 2017, 4, 18. doi:10.3390/cosmetics4020018.
- Yoon, J. Y.; Kwon, H. H.; Min, S. U.; Thiboutot, D. M.; Suh, D. H. Epigallocatechin-3-Gallate improves acne in humans by modulating intracellular molecular targets and inhibiting P. acnes.J. Invest. Dermatol. 2013, 133, 429–440. doi:10.1038/jid.2012.292
- International Standard. ISO 10993-5:2009(E); Biological evaluation of medical devices—Part 5: Tests for In vitro cytotoxicity. Available online: https://www.iso.org/standard/36406.html (accessed on 26 June 2022).

Round 2
Reviewer 1 Report
Authors have addressed all comments, hence the paper is now acceptable for publication in Molecules.
Reviewer 2 Report
I accept introduced corrections.
This manuscript is a resubmission of an earlier submission. The following is a list of the peer review reports and author responses from that submission.
Round 1
Reviewer 1 Report
The manuscript molecules-1718136 entitled ” Kaempferia parviflora Rhizome Extract as Potential Anti-Acne Ingredient” attempts to bring insights into the chemical characterization of Kaempferia parviflora rhizomes and its clinical application as an anti-acne agent.
The study is novel, the idea is original and the obtained results seem to support the authors hypothesis.
Still, the manuscript needs major improvements before publication in Molecules.
I have several comments on the manuscript, as follows:
Introduction Section
Page 1, please rephrase ” Kaempferia parviflora Wall. ex Baker (black ginger). It is one of interesting Thai herbs which is well known and used widely in medical fields due to high antioxidant potency [1-2].”
Please clarify the correlation between the methoxylated flavones identified in Kaempferia parviflora rhizomes with the antimicrobial and anti-inflammatory effects of their derived extracts, as you also used one such constituent (5,7-dimethoxyflavone) as marker compound for the tested extracts
Page 2, line 55 - Please rephrase ”The function of skin is to protect the body from harmful substances.”
Please rephrase and be more concise regarding the aim of the study - Page 2, lines 65-67 ” Therefore, the present study aims to optimize condition of active extract and clinical investigation into its safety, anti-inflammatory, and anti-acne efficacies for possible application in anti-acne products. ”
Results Section
Please be consistent with the abbreviation of the tested extracts in the tables (e.g. in Table 1 there is only mentioned the solvent, in Tables 2, 3 and 5 solvent followed by ext.).
There are different ways of expressing the results: in TPC and TFC the results are expressed per g of extract, meanwhile the antimicrobial and anti-inflammatory activities are expressed per mL extract – Please clarify this aspect
Discussion Section
Please elaborate more the discussion of clinical study results and their relevance.
Material and Methods:
Determination of Minimum Inhibition Concentration:
-please mention the concentration range of tested extracts and positive controls
-please mention the solvent used for the preparation of stock solutions for the extracts and positive controls. Also, if DMSO was used as solvent, please mention its concentration in the used dilutions for antimicrobial testing
-please mention whether you used sterility and growth controls
In Vitro Anti-Inflammatory Activity
-please mention the concentration range of tested extracts and positive controls
-please mention the solvent used for the preparation of stock solutions for the extracts and positive controls
Page 11, Lines 313-314, please rephrase: ” Vancomycin and Gentamicin were used as standard compound as positive control.”
Conclusions:
Please rephrase ” However, further studies should be performed. Additional determinations such as sebum content measurement, and porphyrin content measurement are required to obtain detailed results as well as varying concentrations in the formulation to find minimum concentration which provides the effective results to reduce cost in industry.” And move it in the Discussion section. Please elaborate more the final phrase: ” Overall, black ginger extract could possibly be used as active ingredient in anti-acne product.”
The text needs to be checked by a native English speaker, there are many grammar errors that should be corrected as reading through out the entire manuscript is difficult.
Reviewer 2 Report
L45-47 This sentence should be revised.
L49 There are unnecessary spaces in compound names.
L65 „Therefore, the present study aims to optimize condition of active extract…” This statement is not clear enough.
L76-78 This should be revised. Extract not extraction might contain high amount of phenolics.
L80-84 This part of the manuscript also need to be revised since it is no clear enough.
L94-95 MBC is not presented in Table 2.
L114 The sentence should be reworded.
L115-116 Figure 2 presents chromatograms not parameters mentioned in this sentence.
Figure 1 is very poor quality.
L122 In my opinion „disappear” is not proper word for scientific explanation of extract composition.
L286 The sentence should be reworded.
HPLC analysis conditions should be presented more detailed. There is also no information how the content of DMF was calculated.
Generally, in my opinion the whole manuscript need to be revised by native speaker due to numerous language mistakes.
I think that number of volunteers in the presented study is too low and the composition of gel-cream include too many synthetic ingredients. Moreover, preparing extract with ethyl acetate is not fully safe since that chemical reagent can irritate the skin and repeated contact can cause drying and cracking of the skin.
Reviewer 3 Report
The research named "Kaempferia parviflora rhizome extract as potential anti-acne ingredient" might be of interest to readers, however, the overall presentation is of extremely low quality. The introduction is not clear and precise, especially in supporting the aims of the study. The results section is poorly written with many hard-to-read sentences. The Discussion seems superficial without a clear connection between the results and explanations; this particularly counts for the clinical part of the study. There is also a concern about the cream's applicability in connection with anti-microbial activity and its cytotoxic effect. The authors found that the MIC value in the case of one of the samples is above 3 mg/mL while the cytotoxicity starts from 0.1 mg/mL. So, how does it express its activity in cream if the concentration is lower than MIC? Also, there is no mention of the concentration of the extract which was used in the cream. In the discussion, only the result for NO suppression is commented in the connection with creams efficacy.
English is so bad that it makes this study extremely hard to read.
Based on the above my suggestion is to reject this study.